# Analysis and reporting of adverse events in randomised controlled trials: a review

Rachel Phillips,[1] Lorna Hazell,[2,3] Odile Sauzet,[4] Victoria Cornelius[1]

[1]Faculty of Medicine, School of Public Health, Imperial College London, London, UK
[2]Clinical Research, Drug Safety Research Unit, Southampton, UK
[3]Department of Pharmacy and Biomedical Sciences, University of Portsmouth, Portsmouth, UK
[4]Epidemiologie & International Public Health, Faculty of Health Sciences, Universität Bielefeld, Bielefeld, Germany

**Correspondence to**
Miss Rachel Phillips;
r.phillips@imperial.ac.uk

## ABSTRACT

**Objective** To ascertain contemporary approaches to the collection, reporting and analysis of adverse events (AEs) in randomised controlled trials (RCTs) with a primary efficacy outcome.

**Design** A review of clinical trials of drug interventions from four high impact medical journals.

**Data sources** Electronic contents table of the *BMJ*, the *Journal of the American Medical Association (JAMA)*, the *Lancet* and the *New England Journal of Medicine (NEJM)* were searched for reports of original RCTs published between September 2015 and September 2016.

**Methods** A prepiloted checklist was used and single data extraction was performed by three reviewers with independent check of a randomly sampled subset to verify quality. We extracted data on collection methods, assessment of severity and causality, reporting criteria, analysis methods and presentation of AE data.

**Results** We identified 184 eligible reports (*BMJ* n=3; *JAMA* n=38, *Lancet* n=62 and *NEJM* n=81). Sixty-two per cent reported some form of spontaneous AE collection but only 29% included details of specific prompts used to ascertain AE data. Numbers that withdrew from the trial were well reported (80%), however only 35% of these reported whether withdrawals were due to AEs. Results presented and analysis performed was predominantly on 'patients with at least one event' with 84% of studies ignoring repeated events. Despite a lack of power to undertake formal hypothesis testing, 47% performed such tests for binary outcomes.

**Conclusions** This review highlighted that the collection, reporting and analysis of AE data in clinical trials is inconsistent and RCTs as a source of safety data are underused. Areas to improve include reducing information loss when analysing at patient level and inappropriate practice of underpowered multiple hypothesis testing. Implementation of standard reporting practices could enable a more accurate synthesis of safety data and development of guidance for statistical methodology to assess causality of AEs could facilitate better statistical practice.

## Strengths and limitations of this study

► This is the first review to examine and quantify the methods used for adverse event analysis in randomised controlled trials published in high impact general medical journals.

► This review identifies methodological weakness that need to be addressed as well as good practice that could be adopted.

► Articles included in this review were published in four of the top ranked general medical journals, therefore results are likely to be biased towards better practice.

► Included articles are only for year 2015–2016 and as such may not reflect current practice.

## INTRODUCTION

The methods to analyse and report outcomes to measure benefit from randomised controlled trials (RCTs) are well developed but this progress has not been matched for adverse event (AE) outcomes. An AE is defined as 'any untoward medical occurrence that may present during treatment with a pharmaceutical product but which does not necessarily have a causal relationship with this treatment'.[1] An adverse drug reaction (ADR) is defined as 'a response to a drug which is noxious and unintended …' where a causal relationship is 'at least a reasonable possibility'.[1 2] RCTs provide an opportunity to compare rates of AEs between arms allowing causality to be evaluated. However, contemporary analysis and reporting practices are inadequate.

There are many challenges associated with analysing and reporting AEs in clinical trials. RCTs are typically designed to determine the efficacy of an intervention but are often underpowered to detect important differences in AEs between arms which may suggest an ADR. Often large numbers of AEs are reported during a study, sometimes exceeding the number of patients in the clinical trial. Performing hypothesis tests on these AEs would lead to issues of multiplicity; however, any adjustment for multiplicity would make a 'finding untenable'.[3 4] The use of hypothesis testing may result in the medicinal product being deemed unsafe and a trial being halted too early due to a chance imbalance, or conversely deemed safe and

not stopped early enough resulting in more patients than necessary suffering an ADR.[3 5 6] Unlike efficacy outcomes which are well defined and restricted in number at the planning stage of an RCT, we collect numerous, undefined AEs in RCTs. Furthermore, AE collection requires additional information to be obtained on factors such as severity, timing and duration, number of occurrences and outcome, which for our efficacy outcomes would have all been predefined.

Previous studies have examined the methods for AE collection and presentation only, and highlighted the inadequacies in AE reporting in journal articles.[7–16] In 2004, the Consolidated Standards of Reporting Trials (CONSORT) Group produced an extension to their guidelines for reporting trial results to cover the reporting of harms, however implementation of these guidelines has been shown to be poor.[10 14–17] Recently, a joint pharmaceutical/journal collaboration published practical guidance and examples on what should be reported in journal articles and how it should be displayed to ensure transparency and aid clinical interpretation. They promote the use of clinical judgement in reporting rather than mandatory guidance.[18] While this work has been undertaken there remains uncertainty about practice for reporting and presenting AE data, and in addition the analysis practice for AEs remains a neglected area for review.

The aim of this review was to evaluate contemporary practice for collection, reporting and analysis of AEs in RCTs where the primary outcome was efficacy. The aim being to identify and promote any areas of good practice, while highlighting any areas for improvement.

## METHODS
### Search strategy
The top four general medical journals as ranked by impact factors that publish clinical trials of drug interventions were selected: the *BMJ* (Impact Factor 20.79), the *Journal of the American Medical Association* (*JAMA*, IF 44.41), the *Lancet* (IF 47.83) and the *New England Journal of Medicine* (*NEJM*, IF 72.41). Impact factors quoted are from 2016 to reflect the time period from which the articles were drawn. High impact journals were chosen as we would expect practice in these journals to be of high standard as they include statistical and methodological review. We limited the search to four journals after an initial scoping review revealed around 100 studies would be eligible for inclusion, which was a feasible number to review given the time and resources available and would provide a sufficient number to evaluate practice. One reviewer manually searched the electronic contents table of the journals for reports of original RCTs published between September 2015 and September 2016, inclusive. Any queries regarding eligibility were reviewed and discussed with a second reviewer.

### Selection criteria
The inclusion criteria were phase II–IV RCTs of drug interventions where the primary outcome was efficacy of the intervention. We did not restrict according to number of treatment arms and included both parallel and cluster RCTs. We excluded cross-over RCTs, RCTs with adaptive randomisation, observational studies, case reports, editorials and letters. We also excluded RCTs where the intervention was not a drug product (ie, not classified as a clinical trial of an investigational medicinal product). As the study aimed to assess how the authors report and analyse AEs in studies where the primary outcome was efficacy, trials that were specifically designed to investigate safety as a primary outcome were not included.

### Data extraction
Potentially eligible articles were identified based on titles and abstracts and the full text of these studies were retrieved. Supplementary material was also reviewed if readers were referred here from the main article for further results. Online supplementary table A1 lists all data items captured with guidance given to the reviewers for extraction. The items to be extracted were based on the work by Cornelius *et al* and the CONSORT harm extension with additional items added to capture more specific information on analysis practices.[11 17] Specifically, we focused on the following areas: how AE data were collected (mode of collection, timing) and defined (coding, attribution); how AEs were assessed in terms of severity of the event or relatedness to the medical intervention; if there was any planned AE analysis (final and interim monitoring plans and analysis populations); how events were selected for inclusion in the journal article; how summary event information was presented in the journal article and how AEs were analysed.[11] A more detailed rationale for the choice of items extracted is provided in the online supplementary material table A2.

A data extraction sheet was piloted and then single data extraction was performed by three reviewers (RP, VC and LH) with 10% independent check of a randomly sampled subset to verify quality. Queries were also informally discussed between reviewers on an ongoing basis. Where specific items were flagged for poor agreement these were re-extracted. Any queries during data extraction were shared and disagreements between reviewers were resolved through discussion.

### Data analysis
The proportion of trials reporting each item, 3–4 and 8–34 in online supplementary table A1 were calculated and summary statistics (median and ranges) were calculated for items 5–7. All analyses were performed in Stata V.15.[19] A risk of bias assessment was not undertaken as this study aimed to describe best practice and not evaluate outcomes.

## Patient and public involvement

This review forms part of a wider research project that was developed with input from a range of patient representatives. There were no study participants directly involved in this review but the original proposal and patient and public involvement (PPI) strategy were reviewed by service user representatives (with experience as clinical trial participants and PPI advisors) who provided advice specifically with regard to communication and dissemination to patient and public groups.

## RESULTS

### Data extraction

A total of 585 items were extracted twice across all three reviewers to check the quality of the data extraction. A total of 95 discrepancies were identified. This gave agreement of 84%. During this independent check several items were flagged for potential poor agreement. These items were 100% independently extracted by one author and verified. The items were: study duration; the AE collection method; timing of collection; how binary harm outcomes were summarised; whether continuous outcomes were dichotomised; if continuous outcomes were left as continuous how they were analysed.

### Study characteristics

The search identified 184 eligible trial reports (*BMJ* n=3; *JAMA* n=38, *Lancet* n=62 and *NEJM* n=81) in which a total of 496 911 participants were randomised with a median of 556 participants per trial (range 30, 205 513; IQR 281, 1704). The median trial follow-up was 52 weeks (range 48 hours to 10 years; IQR 24, 104 weeks) and 93% were multicentre trials. Fifty per cent of studies had an active comparator and over 50% of trials received some element of industry funding (table 1).

### Collection and assessment methods

Sixty-two per cent (n=114) of reports made reference to some form of passive (eg, spontaneously reported by patients) AE monitoring or collection methods. Of these, only 46.5% (53/114) or 29% of total reports included specific details (prompts, eg, questions about specific events or AEs in general, questionnaires or diaries) regarding these collection methods (online supplementary table A3, examples 1–2).[20 21] The timing of collection was well documented (91%, 48 out of 53 reports) in the reports that included specific details about the prompts used to collect AEs. Although specific details on clinical examinations (eg, vital signs and blood pressure) and laboratory tests were not widely reported (only 57% of reports (95 out of 166 reports with clinical examinations and/or laboratory results presented) included details on the timing of such assessments), it was often clear from the results presented that participants had undergone these assessments (83% and 79% of studies reported clinical and laboratory results, respectively) (table 2).

**Table 1** Characteristics of included studies

| Characteristic | n=184 | | |
| --- | --- | --- | --- |
| | Median | (IQR) | Min, max |
| Sample size | 556 | (281, 1704) | 30, 205 513 |
| Centres* | 35 | (12, 100) | 1, 1368 |
| Trial duration (weeks)† | 52 | (24, 104) | 0.3, 521 |
| | n | % | |
| Journal | | | |
| *BMJ* | 3 | 1.6 | |
| *JAMA* | 81 | 44.0 | |
| *Lancet* | 38 | 20.7 | |
| *NEJM* | 62 | 33.7 | |
| Funded by | | | |
| Public | 70 | 38.3 | |
| Industry | 80 | 43.7 | |
| Both | 33 | 18.0 | |
| Centre | | | |
| Single centre | 12 | 7.0 | |
| Multicentre | 161 | 93.0 | |
| Control | | | |
| Placebo | 95 | 51.6 | |
| Active | 80 | 43.5 | |
| Both | 8 | 4.4 | |
| Neither‡ | 1 | 0.5 | |

*Eleven reports did not specify the number of centres.
†Two reports did not specify trial duration.
‡One trial compared interventional drug to behavioural change intervention.
*JAMA*, Journal of the American Medical Association; max, maximum; min, minimum; *NEJM*, New England Journal of Medicine.

### Prespecified analysis

Thirty-one per cent of reports provided information on the planned analysis for AEs in the statistical analysis section of the paper and 45% prespecified a safety population (online supplementary table A3, examples 3–4 and table 2).[22 23] A quarter of trials reported planned interim analysis with stopping criteria (table 2), five (2.7%) of which included specific criteria on stopping for a harmful event (online supplementary table A4).[24–28]

### Selection of AEs and reporting practices

Two reports only made generic statements regarding AE data: '*there were no significant adverse events related to the procedure*' and '*no excess in mortality or major adverse events were found…*'. Three reports made no mention of AEs throughout the manuscript.[29–33]

Twenty-four (13%) trials only provided a summary of the number of AEs or serious AEs rather than listing the actual AEs that occurred. For example, '*Six serious adverse events occurred in the acetaminophen group and 12 in the*

**Table 2** Collection, assessment and analysis methods reported by studies

| Section | Component | Data item | n=184 n | % |
|---|---|---|---|---|
| **Collection** | | | | |
| | How was AE/harm information collected? | | | |
| | | Passive collection | 114 | 62.0 |
| | | Prompted collection (n=114) | 53 | 46.5 |
| | | No method of collection reported | 70 | 38.0 |
| | Did they undertake proactive screening? | | | |
| | | Clinical examinations | 153 | 83.2 |
| | | Laboratory tests | 146 | 79.4 |
| | Timing of prompted collection specified (n=53) | | 48 | 90.6 |
| | Timing of active collection specified (n=166) | | 95 | 57.2 |
| | Which, if any, dictionary was used to code AE data? | | | |
| | | CTCAE | 18 | 9.8 |
| | | MedDRA | 43 | 23.4 |
| | | CTCAE and MedDRA | 1 | 0.5 |
| | | DAIDS | 2 | 1.1 |
| | | ICD-10 | 1 | 0.5 |
| | | Researcher defined | 2 | 1.1 |
| | | Other | 3 | 1.6 |
| | | No dictionary reported | 114 | 62.0 |
| **Assessment** | | | | |
| | Who assigned attribution to study drug? | | | |
| | | Blinded assessor | 9 | 4.9 |
| | | Unblinded assessor | 7 | 3.8 |
| | | Both | 1 | 0.5 |
| | | Not specified | 164 | 89.1 |
| | | Not applicable* | 3 | 1.6 |
| **Analysis** | | | | |
| | Was any analysis for AEs specified in the methods section? | | | |
| | | Yes | 57 | 31.0 |
| | Was a population for AE analysis specified? | | | |
| | | Yes | 82 | 44.6 |
| | Was there a planned interim analysis with stopping criteria? | | | |
| | | No | 138 | 75.0 |
| | | Yes for efficacy | 24 | 13.0 |
| | | Yes for efficacy and futility | 11 | 6.0 |
| | | Yes for efficacy and safety | 3 | 1.6 |
| | | Yes for efficacy, futility and safety | 2 | 1.1 |
| | | Yes but no other details given | 6 | 3.3 |

Denominator specified in item column if it differs from total sample.
*Three reports made no reference to AE data throughout the article.
AE, adverse event; CTCAE, Common Terminology Criteria for Adverse Events; DAIDS, The Division of AIDS; ICD-10, International Classification of Diseases 10th revision; MedDRA, Medical Dictionary for Regulatory Activities.

*ibuprofen group*'.[34] Of these 24 trials, 10 did provide specific details of the types of events in an appendix. This means 8% of trials either did not report AEs or only included a summary (table 3).

Eighty-nine per cent of trials reported a subset of all the AEs they collected. How AEs are 'selected' for inclusion in the article was not consistent or clear, and in 3% of studies it was impossible to discern how the authors had selected the AEs they presented for inclusion. Twenty-six per cent of reports selected events based on a frequency threshold, for example, events experienced by >x% in any group; 9% of reports used a measure of severity to

**Table 3** Summaries of results presented by studies

| Component | Data item | n=184 n | % |
|---|---|---|---|
| **What was reported in the manuscript?** | | | |
| | Actual AE terms | 73 | 39.7 |
| | Summaries of AE type (eg, AE, SAE) | 24 | 13.0 |
| | Both | 80 | 43.5 |
| | Neither | 7 | 3.8 |
| **What was reported in the appendix?** | | | |
| | Actual AE terms | 76 | 41.3 |
| | Summaries of AE type (eg, AE, SAE) | 7 | 3.8 |
| | Both | 22 | 12.0 |
| | Neither | 3 | 1.6 |
| | Not applicable* | 76 | 41.3 |
| **Which population was the AE analysis performed on?** | | | |
| | All randomised | 54 | 29.4 |
| | Those that took at least a single dose | 75 | 40.8 |
| | Other | 35 | 19.0 |
| | Not specified | 17 | 9.2 |
| | Not applicable† | 3 | 1.6 |
| **Were drop-outs/withdrawals reported?** | | | |
| | No | 33 | 17.9 |
| | Yes by treatment arm | 144 | 78.3 |
| | Yes overall | 2 | 1.1 |
| | Not applicable‡ | 5 | 2.7 |
| **Were withdrawals due to AEs reported? (n=146)** | | | |
| | No | 89 | 61.0 |
| | Yes | 51 | 34.9 |
| | Not applicable§ | 6 | 4.1 |
| **Were specific AEs causing withdrawals reported? (n=51)** | | | |
| | No | 39 | 76.5 |
| | Yes | 12 | 23.5 |
| **How were binary AE outcomes summarised by arm?** | | | |
| | Not summarised¶ | 6 | 3.3 |
| | Number of people with an event | 154 | 83.7 |
| | Number of events | 11 | 6.0 |
| | Both | 12 | 6.5 |
| | Unclear | 1 | 0.5 |
| **Were frequencies of AEs reported by arm?** | | | |
| | No | 5 | 2.7 |
| | Yes for some | 13 | 7.1 |
| | Yes for all | 160 | 87.0 |
| | Not applicable¶ | 6 | 3.3 |
| **Were percentages of AEs reported by arm?** | | | |
| | No | 18 | 9.8 |
| | Yes for some | 25 | 13.6 |
| | Yes for all | 135 | 73.4 |

Continued

| Table 3 Continued | | | |
|---|---|---|---|
| | | **n=184** | |
| **Component** | **Data item** | **n** | **%** |
| | Not applicable¶ | 6 | 3.3 |
| Were between arm differences and 95% CI of AEs reported? | | | |
| | No | 141 | 76.6 |
| | Yes for some | 18 | 9.8 |
| | Yes for all | 19 | 10.3 |
| | Not applicable¶ | 6 | 3.3 |
| Were statistical significance tests between arms on AEs reported? | | | |
| | No | 92 | 50.0 |
| | Yes for some | 31 | 16.9 |
| | Yes for all | 55 | 29.9 |
| | Not applicable¶ | 6 | 3.3 |
| Were continuous AEs outcomes dichotomised for summaries? | | | |
| | No | 10 | 5.4 |
| | Yes for some | 28 | 15.2 |
| | Yes for all | 108 | 58.7 |
| | Not applicable | 38 | 20.7 |
| If continuous outcomes were left as continuous what between arm analyses was performed? (n=38) | | | |
| Differences in measures of central tendency estimated with 95% CI | | | |
| | No | 23 | 60.5 |
| | Yes for some | 1 | 2.6 |
| | Yes for all | 14 | 36.8 |
| Between arm hypothesis tests performed | | | |
| | No | 12 | 31.6 |
| | Yes for some | 2 | 5.3 |
| | Yes for all | 24 | 63.2 |
| Were any 'signal detection' approaches used? | | | |
| | No | 184 | 100.0 |
| | Yes | 0 | 0.0 |
| Were there any graphical presentations of AE outcomes? | | | |
| | No | 162 | 88.0 |
| | Yes | 22 | 12.0 |
| Were summaries of severity rating of AEs reported? | | | |
| | No | 103 | 56.0 |
| | Yes for some | 41 | 22.3 |
| | Yes for all | 35 | 19.0 |
| | Not applicable** | 5 | 2.7 |
| Were number of SAEs reported? | | | |
| | No | 44 | 23.9 |
| | Yes overall | 2 | 1.1 |
| | Yes by treatment arm | 132 | 71.7 |
| | Not applicable†† | 6 | 3.3 |
| For SAEs was relatedness given? (n=134) | | | |
| | No | 77 | 57.5 |
| | Yes for some | 18 | 13.4 |

**Table 3** Continued

| Component | Data item | n=184 | |
|---|---|---|---|
| | | n | % |
| | Yes for all | 38 | 28.4 |
| | Yes overall | 1 | 0.8 |
| Were there any AEs where information on duration of events was reported? | | | |
| | No | 175 | 95.1 |
| | Yes | 9 | 4.9 |
| Were there any AEs where information on the time of occurrence of events was reported? | | | |
| | No | 132 | 71.7 |
| | Yes | 52 | 28.3 |
| If any significance tests were performed on AEs was multiplicity of events accounted for? | | | |
| | No | 81 | 44.0 |
| | Yes | 3 | 1.6 |
| | Not applicable | 100 | 54.4 |
| Did the report reference the CONSORT extension to harms? | | | |
| | No | 184 | 100.0 |
| | Yes | 0 | 0.0 |

*Make no reference to the appendix.
†Three reports made no reference to AE data throughout the article.
‡Five reports indicate no withdrawals.
§Six reports specify the number of withdrawals and reasons but none of the reasons are related to AEs.
¶This includes three reports with no AE data (as per footnote †), two reports that provide generic statements regarding AE data and one report that only reported continuous outcomes.
**This includes three reports with no AE data and two reports that provide generic statements regarding AE data (as per footnote ¶).
††Six papers specifically state that no SAEs occurred.
AE, adverse event; CONSORT, Consolidated Standards for Reporting Trials; SAE, serious adverse event.

select events, for example, AEs of grade 3 or higher; 23% of reports included events based on seriousness and 8% included AEs based on relatedness to treatment (percentages are not independent as the majority of reports used several different criteria for selection). Online supplementary tables A5 and A6 provide full details of selection criteria used.

We found that 41% of trials analysed AEs in participants that received at least one dose, 29% of trials used all randomised participants and 9% did not specify the analysis population (table 3). Further details on analysis populations used are given in online supplementary table A7.

Nearly 80% of trials reported the number of participants who withdrew from the trial; of these 35% (51 of 146 reports) reported whether the withdrawals were due to AEs and of these 24% (12 of 51 reports) reported the actual events that caused withdrawals. Results presented and analysis performed was predominantly on 'patients with at least one event' with 84% of reports providing no information on the number of events occurring. An example of how to incorporate information on number of events is presented in the study by Lind *et al.*[35] Forty-one per cent of trials reported information on the severity of AEs. Five per cent of trials included a report of at least one event with duration, but presenting such data is limited in

the main report. The trials that did present this information did so in a variety of ways. For example, incorporating the information into the AE table with summary statistics such as the mean duration of certain events or presenting it for a subgroup of events in the footnotes of AE tables, for example, '*One event of non-serious squamous cell carcinoma (day 210, resolved on day 215; adalimumab treatment was not interrupted)*'.[36–38] Twenty-eight per cent of reports included information on the timing of AEs (table 3).

Serious AEs were typically well documented (73%) and six reports (3%) explicitly stated that no serious events had occurred. However, for 44 reports (24%) it was not possible to discern if no serious events had occurred or whether they were simply omitted from the report. Forty-two per cent (57 of 134 reports) of reports included details on whether the events had been classified as related to the intervention (table 3).

## Analysis of AE outcomes

The majority of trials summarised binary outcomes using frequencies (94%) and percentages (87%). Despite a lack of power to undertake formal hypothesis testing, 47% reported p values for binary outcomes. For example, '*There were no between-group differences in the rate of patients with at least one adverse event (16.7% (14 patients) in the clopidogrel group vs 21.8% (19 patients) in the placebo group;*

*difference, –5.2% (95% CI –17% to 6.6%); p=0.44)'.* However, with a total safety population of 171 such a test would have only had 13% power to detect such a difference and was therefore substantially underpowered. The conclusion that '*No significant increase in adverse events was observed*' makes no reference to the 95% CI presented which indicates that the findings were in fact compatible with a 17% decrease in experiencing at least on AE as well as a near 7% increase.[39]

There was a pervasive practice (59%) of categorising continuous clinical and laboratory outcomes. Of the trials that did not dichotomise continuous AE data, nearly 70% performed some form of statistical significance testing (table 3). While continuous outcomes do not suffer to the same degree regarding lack of power, multiple testing is still a problem; however, no multiplicity corrections for continuous outcomes were performed.

Of the trials that performed statistical significance testing on AE data, only three made an adjustment for multiplicity of tests (all three on dichotomised outcomes).[36 40 41] Two of which used a Bonferroni correction and adjusted for the number of pairwise comparisons between each of the treatment groups for each individual event rather than the total number of significance tests performed. As such, both analyses would have still been affected by issues of multiple testing.

Twelve per cent of reports used graphs to illustrate AE data (table 3). The CONSORT extension highlighted the value of graphs for summarising such data, especially for conveying information on time-to-event outcomes.[42] An example of such a plot is included in the study by Lingvay *et al*[43] (see supplementary eFigure2 of reference 43).

We assessed any reference to the CONSORT harm extension and found that none of the included studies mentioned it. Of the four journals included in the review, the *Lancet* was the only journal that made specific reference to the harm extension in their guidelines to the authors.

## DISCUSSION

The safety profile of a medicinal product is established through evidence collected from several sources including clinical trials, observational studies and spontaneous reports.[44] The advantage of clinical trial data is that these provide a controlled comparison of the rate of AEs allowing causality to be evaluated but have the disadvantage that the sample size is often not large enough to detect rare ADRs.

To ensure that a useful and comprehensive picture of the safety profile is provided to all relevant parties clear reporting of AEs from clinical trials is required. Recent research has shown the quality of reporting is substandard.[7–16] The aim of this study was to review contemporary practice across four leading medical journals for AE collection, reporting and analysis practices, highlighting any areas for improvement and examples of good practice. We found that the collection, reporting and analysis

of AE data in clinical trials is inconsistent and RCTs as a source of safety data are underused. Analysis of AE was often inappropriate with suboptimal practice including ignoring valuable information on repeated events and inappropriate practice of underpowered multiple hypothesis testing.

## Collection and assessment methods

The CONSORT extension to harm was developed with the aim to improve reporting of safety data in RCTs.[42] None of the included studies referenced the CONSTORT harm extension and of the items in our review that are covered in CONSORT many were not well reported.[17] This suggests that the CONSORT extension is not being routinely adopted by the authors to aid their reporting. Most journals now request that the authors include a completed CONSORT checklist when they submit their article but we are not aware of any journal that request the CONSORT harm extension to also be submitted. Of the four journals in this review, the *Lancet* is the only journal that makes specific reference to the harm extension in their guidelines to the authors. The CONSORT statement contains a single item related to safety, item 19: 'all important harms or unintended effects in each group' should be reported.[42] This may explain why some items listed on the CONSORT extension for harm were reported by so few trials. The mandatory submission of CONSORT harms by journals may support better reporting.

We found that the method of AE collection was poorly reported. This has important implications for the type and frequency of AEs reported with 'passive collection resulting in fewer recorded AEs'.[45 46] Where the method was given the timing of collection was typically also reported and we would recommend continuation of this practice. The frequency of AE collection has further important implications on the number of events reported. More frequent assessment and longer follow-up will result in more AEs reported.[17] It is important to consider these factors when making conclusions about the safety profile.

The method of attribution between drug and AE was another area where reporting practice was inadequate. However, the joint pharmaceutical/journal collaboration indicate that such attribution has 'limited value' given the 'inherent subjectivity in such attribution'.[18]

## Prespecified analysis

We found that formal assessments of AEs regarding stopping for emerging ADRs using statistical rules was rare. Subjective assessments of overwhelming amounts of data could easily lead to potential signals of harm being missed. There could be benefits to incorporating more objective statistical methods alongside clinical review to assist the evaluation of AE information to help better identify drug harm relationships. Graphical displays have gone some way towards aiding interpretation.[47–51]

## Selection of AEs and reporting practices

Due to space constraints in journal reports AE information is often included in the appendix. While we encourage use of

appendices and supplementary material for including additional detail on AEs, we caution the authors against depositing all AE data into such documents without attempting to present a summary of the AE profile in the main article. It is important that the main report strikes a balance between efficacy and harm therefore allowing a risk-benefit assessment to be made solely from the article.

The failure to report any information on AEs restricts interpretation and prevents a risk-benefit assessment. We identified two reports that made generic summaries of the overall safety profile and it was clear in both that there had been harmful effects. However, the authors did not include any further information. Three reports contained no information leaving readers uninformed as to any additional information these studies may provide on the safety profile. Ambiguous reporting prevents building an accurate picture of the safety profile. As such profiles are developed on accumulating evidence, it is important that each study report to the same standard and information is not wasted.

We found that the selection criteria used by the authors to decide what AEs to include in the report were arbitrary and inconsistent. This will have important implications when synthesising data across studies to construct safety profiles. Authors would benefit from guidance to facilitate consistency but research in this area is lacking. Lineberry *et al* recommended clinically relevant events that should always be reported (deaths, serious AEs and events leading to discontinuation of intervention) and criteria that should be considered when deciding what other AEs to report, for example, interest based on the disease(s) under investigation, comorbidities of the study population, intervention mechanism, trial duration.[18] Standard outcomes for a drug class would be one potential solution to avoid issues of inconsistency suggested by Cornelius *et al*.[11]

CONSORT recommend that AE analyses should be performed on the intention-to-treat (ITT) population to maintain the random assignment.[17] However, it is clear from our review that this population label is not always appropriately and consistently applied. There is a tendency for studies to make modifications to the ITT population. Using the ITT or modified ITT population is likely to underestimate the risk by inflating the denominator with participants who may have never received the study drug.[52] Such estimates are appropriate for health economic evaluations where estimates of the cost-effectiveness will inform policy level decisions regarding how to treat the population. However, a more appropriate population for AE analysis to inform prescriber and patient decisions may be those that receive at least one dose. It is important that the authors clearly define and specify a suitable safety analysis population and consider how this affects their conclusions.

Proxy outcomes can be used as a measure of the impact of AEs on patients. Examples include the number of withdrawals due to any reason, withdrawals due to AEs, the number of events an individual experiences, the severity of the AE and the duration. A high proportion of trials reported withdrawal for any reason and this is likely to be as a result of the CONSORT recommendations.[42] The other outcomes were not frequently reported and increasing this could facilitate interpretation.[17] This information would permit better evaluation of the impact of AEs and the tolerability of the intervention to inform patients' and clinicians' treatment decisions. Reporting numbers that experience at least one event only and not providing information on repeated events masks valuable information that may be important to the patient and the cost-effectiveness evaluation. For example, chronic, repeated headaches over an extended duration will have an important impact for patients compared with a single headache or headaches over a short duration but it is not possible to distinguish between these two scenarios when reported as 'at least one event'.[18] Severity of events was also an important aspect that was often not differentiated. For example, there would be a different impact on patients' quality-of-life with mild compared with severe nausea, which could lead to changes in dosing regimens. Displaying such information for all AEs in tables would soon become overwhelming and make interpretation difficult. Graphical approaches have been suggested as a solution to aid review. Examples of such a plot can be found in UW–Madison SDAC.[53] Online appendices and supplementary material provide more opportunity to include this important information.

**Table 4** Recommendations to improve adverse event analysis and reporting in clinical trial report publications

|  | Recommendation |
| --- | --- |
| Analysis | Incorporate objective statistical methods to assist the evaluation of adverse event information. |
|  | Consider avoiding dichotomising continuous data. |
|  | When count outcomes are available (such as repeated events within participants) use appropriate statistical methods. |
|  | Clearly define exposure and specify a suitable safety analysis population. |
|  | Use graphical approaches to help summarise large amounts of data. |
| Reporting | Report adverse event data according to the CONSORT harm checklist. |
|  | Increase the uptake of mandatory submission of CONSORT harm by journals. |
|  | Include a relevant summary of the adverse event profile in the main article. Resist depositing all adverse event data into appendices without summarising. |

CONSORT, Consolidated Standards for Reporting Trials.

For serious AEs information on the time of likely onset can be useful information to inform patient monitoring plans. For example, the documented risk of suicide and suicidal ideation within the first few weeks of starting an antidepressant allows patients and prescribers to remain alert and monitor closely for this period. Nearly a third of reports included such information and we would encourage the authors to adopt this practice.

### Analysis of AE outcomes

The majority of trials in this review included a balanced report of AEs alongside benefit. However, many included generic statements regarding the safety profile such as 'the intervention was well tolerated' or 'the intervention exhibited a good safety profile' and these were frequently based on post hoc statistical tests. Guidelines caution against such tests.[18] The results of which are difficult to interpret as a lack of significance does not indicate that the intervention is safe and conversely multiple testing without adjustment will increase the number of significant differences due to chance.[54 55]

Graphs are an efficient method to convey and interpret large amounts of data and can make it easier to flag potential safety signals.[50 51 53] Twelve per cent of studies included in the review used graphs to present AE data and an example of one such report is given in the study by Coovadia et al[56] (see online supplementary eTable of reference 56).

Recommendations for consideration for immediate adoption by the clinical trial community are summarised in table 4.

### Limitations of trials

Trials are a valuable source for high-quality AE data but compared with observational studies have smaller sample size, follow-up periods and generalisability, which restrict the ability to detect rare ADRs, ADRs with long latency and drug interactions in complex populations. The typical duration of a trial means there is often insufficient follow-up to fully characterise the safety profile as it provides limited information on long-term exposure. Stringent inclusion criteria restrict the population the intervention is assessed in and so limited information on drug interactions is obtained.[5]

### Limitations of this study

Articles included in this review were published in four of the top ranked medical journals, therefore results are likely to be biased towards better findings than we would expect if we included all RCTs. Articles are only for year 2015–2016 and as such may not reflect current practice. We also acknowledge that only completing 10% independent check of extracted data would not have removed subjectivity from the data extraction but are happy that ongoing discussion between the authors to clarify any queries would have kept this to a minimum. Despite these limitations, this review characterises what those leading the field are doing and provides some examples of good practice that could be adopted.

### Conclusions and recommendations for future work

RCTs are a valuable source of information when establishing the safety profile of medicinal products. Our review has demonstrated that data are not being fully used. Analysis of AE data is frequently inappropriate and RCT reports published over a recent period in high impact general medical journals often provide insufficient and inconsistent information to allow a comprehensive summary of the safety profile to be established.

This research has identified two areas that would benefit from future research: (i) improving the consistency of reporting important AE outcomes across trials to facilitate comparison and synthesis. This is in line with work from the COMET Initiative group (http://www.comet-initiative.org/) and the development of CORE safety outcomes by drug class could be considered.[7] (ii) Evaluation of methods to analyse AEs in RCTs.

**Contributors** RP conceived the idea for this review, conducted the search, carried out data extraction and analysis and wrote the manuscript. VC conceived the idea for the review, performed data extraction, critical revision of the manuscript and supervised the project. LH performed data extraction and critical revision of the manuscript. OS performed critical revision of the manuscript.

**Funding** Rachel Phillips is funded by a National Institute for Health Research (NIHR) Doctoral Fellowship.

**Disclaimer** This paper presents independent research funded by the National Institute for Health Research (NIHR). The views expressed are those of the author(s) and not necessarily those of the NHS, the NIHR or the Department of Health and Social Care.

**Competing interests** None declared.

**Patient consent for publication** Not required.

**Provenance and peer review** Not commissioned; externally peer reviewed.

**Data sharing statement** Data are available upon reasonable request made to the corresponding author.

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
