## [Reviewer comments · BMJ Open]

ARTICLE DETAILS

TITLE (PROVISIONAL)	Analysis and reporting of adverse events in randomised controlled trials: a review
AUTHORS	Phillips, Rachel; Hazell, Lorna; Sauzet, Odile; Cornelius, Victoria

VERSION 1 – REVIEW

REVIEWER	Liliane Zorzela University of Alberta, Canada
REVIEW RETURNED	22-Jun-2018

GENERAL COMMENTS	I read with interest the manuscript: Lessons to learn on the reporting of AEs in RCTs. A SR. This is a very relevant topic and the authors identified important information that should be voiced. I have a few concerns with the methodology and reporting of findings for this paper and I'll summarize in a few lines here and in more depth below. 1- I suspect this manuscript is NOT a systematic review (SR). SRs should follow a very systematic method to assure no relevant data has been overlooked. The authors do not report how screening of titles and full papers were done (if in duplicate or not, so I suspect not) and the data collected was not double checked (only 10% of data extracted was checked for accuracy). Also, there is no mention on ROB assessment, it may considered to be omitted in this review, but the authors have to mention the reasons to omit. I believe this paper should be classified as a rapid review based on the above, unless authors can document that many mandatory steps for a SR were indeed followed. In more detailed: Strengths in limitation box: Item 1: change to rapid review Item 2- can you also mention that many weaknesses in methods of reporting were identified and have to be addressed. Not that you only identified good examples. Item 4- I don't think your review addressed this. Did you specifically review the guidance for best statistical methods to assess causality in RCTs? If you just got a pool of studies form 4 journals and published within 1 year you can not make such statement. 1. Objectives: I suspect the term "best practice" is not accurate. Please use "current practice" or something with more equipoise. Please add that you only looked in RCTs of efficacy. Attention to these changes at the discussion section as well. 2. Search strategy:
---

	Minor: list journals in alphabetical order 3. Methods 3.1 Screening? Authors do not report how screening was done. Screening is an important step in a SR and must be clearly reported based on the PRISMA guideline (item 9- study selection) -please report. Screening and data extraction are key factors for a SR, assuring no data is missing. 3.2Data extraction: Only 10% of data was verified by a second reviewer. I believe since the data extraction could be quite extractor dependent, due heterogeneity in format of AE reports in RCTs, the data extraction may require some degree of 'judgement call' and it would be more appropriate to have all data checked by a second reviewer, as it is recommended/desirable in a SR. If this step was not followed, I would suggest describing your review as a rapid review, where the reader understands that some main steps form a SR process were omitted to optimize time and therefore some flaws in the process are expected. Please move to results the information that: Agreement between authors was over 80% and please add a little more information on this): How many items had to be re-extracted... 3.3Data analysis Please justify why different methods of data analysis were chosen. 4. Results: Table 3: How AEs info was collected? a) The percentages and N do not add up- did studies used more than one methods of AE assessment (active and passive?) if yes, please report it. How many did not report any of these? Please add this info too The non applicable item for attribution say that 3 reports included no AE data. What does it mean? No AE were identified/ were not reported/were reported as occurred but no further explanation? Please be clear. Selection of AEs and reporting practices- Here your say that 5 trials did not report any information on AEs and in table 3 you say 3 trials. Please check what is correct. Discussion In the discussion you have a subtitle called "principle findings" but none of the things reported here were found in your review. Please re-write (or removing the subtitle or adapting the content to our findings). Use of CONSORT Harms under principal findings: You have not specifically checked for the use of CONSORT Harms, so you cannot say that consort harms has not been used based on your findings. If you specifically looked for CONSORT Harms items, please say it earlier in the manuscript (methods and results). Maybe I missed, but I did not see if you looked how many journals endorse the CONST Harms. You should not present new information in the discussion (it should belong to results). If I missed the description in results I apologize.
--	--

	Please add as a limitation that your study was also limited to 2015-2016 and this may not reflect most current practices and previous years.
--	--

REVIEWER	Reidar P. Lystad Macquarie University, Australia
REVIEW RETURNED	24-Jun-2018

GENERAL COMMENTS	Thank you for the opportunity to review the manuscript titled “Lessons to learn from the reporting of adverse events in randomised controlled trials: a systematic review”. The manuscript is relatively well written and it presents interesting information about adverse event reporting in RCTs published in four leading medical journals. However, I have several comments/suggestions.  1. The rationale for this review could be made clearer in the Introduction section. The authors appear to cite a number of previous studies documenting the inadequacies of AE reporting in journal articles, but it is not made explicitly clear what the gaps in this literature actually are. The first bullet point in the Strengths and Limitations indicates that the unique contribution of the present review is its examination of analysis practices for AEs in RCTs. This seems reasonable, but if so then the present review ought to focus more on this particular aspect. Perhaps this should be reflected in the title as well. 2. There is no strong justification for why only four journals were searched. I understand that the four journals are among the leading general medicine journals, but what was the criteria for selecting journals. Given that only four journals are used as sources, I am not sure it is appropriate to call this a ‘systematic review’. At best it is an extremely restrictive systematic review. 3. Similarly, what is the justification for only including RCTs from the 13-month period between September 2015 to September 2016, inclusive? Why 13 months? Moreover, if the present review is supposed to examine current best practices, then why not include more recent RCTs, say, from 2017? 4. It is unclear whether the study selection process was conducted by one or more independent reviewers. If it was conducted by two or more independent reviewers, then what was the level of agreement between reviewers? And how was potential disagreement resolved? 5. In regard to data extraction, the authors state that agreement between reviewers was over 80%. Please state the exact figure. How was the level of agreement measured? Cohen’s kappa? Intraclass correlation coefficient? 6. The Results section is unreasonably long (i.e. a whole 10 pages!). As mentioned above, it would be helpful if the authors focused on the unique contribution of their review, that is, the reporting practices of analysis of AEs. (Much of the other information could be relegated to appendices if necessary.) I would have liked to see a little bit more detail on the reporting practices of AEs. How many RCTs were powered to detect differences in AEs? How many inadequately powered RCTs (inappropriately) conducted hypothesis tests of differences in AEs?
---

	7. The limitations of the present review are understated. Restricting the source of RCTs to four leading medical journals will of course have a major impact on the generalisability of the findings of the present review. But are there no limitations other than the included RCTS being biased toward better results because they were published in top ranking medical journals? 8. One issue I find problematic is that the authors begin with the assumption that RCTs from the four leading medical journals will inform us of current best practice (i.e. lessons to learn), then they proceed with a purely descriptive synthesis of these practices. So far so good. But then the authors turn around, instead of assuming these are best practices, they highlight ways in which these practices are inadequate or inappropriate by pointing to yardsticks such as the CONSORT Statement. At the very least the authors should make it clear up front (i.e. in the Methods section) what criteria or standard the reporting practices are evaluated against.
--	--

REVIEWER	Martin Howell University of Sydney, School of Medicine
REVIEW RETURNED	26-Jun-2018

GENERAL COMMENTS	Overall comments: This review adds to the number of systematic reviews on reporting of adverse events in RCTs. The findings demonstrate that despite the CONSORT harms extension having been published in 2004 there has been no apparent improvement in reporting. In short it is difficult to draw any meaningful conclusions on AEs in evidence from RCTs, be they long term serious outcomes or short term drug related side effects. The review highlights again the key reporting deficiencies and to this extent does not provide any new knowledge, however it confirms the pervasive and persistent nature of the inadequate reporting of AEs. Specific comments:  Given that there have been a number of systematic reviews, that have used similar methods for appraisal and synthesis, it is not reasonable to say that this is the first SR to examine and quantify analysis practices for AEs in RCTs. Publication of an RCT in one of the selected journals does mean that the RCT represents 'best practice'. Rather this would be assessed on the basis of study design and risk of bias which has not been undertaken. Indeed it could be argued that they do not represent best practice due to the deficiencies in reporting of AEs. I agree with the authors that assessment of risk of bias and 'quality' of individual trials is not relevant to the objective of the review. I suggest that reference to best practice be removed. The important point is that the RCTs that have been published over a recent period in examples of high impact general medicine journals are deficient. The limitation of the study is not that these might be 'better' trials, but that the review is limited to four journals and thus provides a 'snap shot' rather than a systematic assessment. A further limitation is that the review is of single publications, whereas AE data may be over a number of reports and would need a systematic review conducted at a study level to capture. Why were trials designed to investigate safety excluded as these may be reports from efficacy studies.
---

	4. Table 2 could be removed as it does not provide sufficient detail for the reader to be able to form a judgement as to how representative o appropriate or otherwise of the four examples. 5. The authors should consider structuring the assessment more in line with the 10 domains and 23 items of the CONSORT harms extension. This could be employed as a simple checklist to highlight areas of deficiency as has been done in other reviews with results shown graphically in a manner similar to risk of bias assessments. 6. The rationale for the components and data items shown in Tables 3 and 4 are not clear this should be expanded in the methodology. 7. The review has not presented an analysis of the types of AEs reported and whether they are 'important' or whether they would be expected to be associated with the intervention. For example other reviews have shown a miss-match between known drug related side effects and AEs reported in RCTs. This does not point to a need for research on outcomes, rather a need for reporting to comply with CONSORT. The question of establishing important outcomes e.g. as per COMET is perhaps warranted for other reasons, but this review shows a problem with reporting. The main problem with the current reporting is that when an AE is not reported, it is not known whether it did not occur, was not included in the outcomes and therefore not measured or recorded, or was measured using unknown or inappropriate methodology. In short AE reporting is unreliable. 8. The authors could suggest key items of AE reporting that if addressed would improve reliability.
--	--

VERSION 1 – AUTHOR RESPONSE

nWe would like to thank the editorial team for their comments and giving us the opportunity to respond and revise our manuscript accordingly.

Comments from the Editorial Team

- 1) We agree with the reviewers who say this isn't a systematic review. It is actually more of a cross-sectional analysis. For example, see a very similar type of paper below:
Differences in reporting serious adverse events in industry sponsored clinical trial registries and journal articles on antidepressant and antipsychotic drugs: a cross-sectional study
<https://bmjopen.bmj.com/content/4/7/e005535>

Response

We thank the editor for their observation regarding the labelling of this study as a systematic review. We do not believe our research meets the alternative definition proposed by reviewer 1 as a rapid review i.e. timely information for decision making. The reason we referred to it as a systematic review is because the method to identify eligible literature and the process of evaluation of these papers was systematic. However we acknowledge that the restrictions imposed on the review might mean it does not fully meet the standard definition of a systematic review. Therefore we suggest that we remove the word 'systematic' and simply call it 'a review' and have amended the manuscript throughout.

Marked revisions

Title: "Analysis and reporting of adverse events in randomised controlled trials; a review"

The word 'systematic' has been removed from the study description throughout the article.

- 2) We didn't feel that the rationale of choosing just these 4 journals was very clear. Can you please clarify this? For instance, JAMA Internal Medicine and Annals of Internal Medicine publish many trials (The BMJ actually doesn't publish that many) and their IF trails behind the Impact Factor of The BMJ. You also need to update the IFs on page 6 since they seem to be the ones from last year.

Response

The aim of this review was to examine current practice in high impact general medicine journals to establish what leaders in the field were doing. Therefore we chose the top four ranked general medical journals that published clinical trials as measured by their impact factor in 2016. We would expect practices in these journals to be of a higher standard than trials published in any journal as they include statistical and methodological review. From experience we knew the standard in general is poor and we were aiming to survey the 'best practice' approaches.

The impact factors quoted in the manuscript were the latest available at the time of submission and also reflect the period from which the articles were drawn. We have now clarified this in the text. We are happy to add the 2017 figures if the editor wishes us to do so? There was also an error in the figure quoted for the Lancet and this has been amended.

Marked revision

"The top four general medical journals as ranked by impact factors that publish clinical trials of drug interventions were selected: The BMJ (Impact Factor (IF) 20.79), the Journal of the American Medical Association (JAMA, IF 44.41), the Lancet (IF 47.8300), and the New England Journal of Medicine (NEJM, IF 72.41). Impact factors quoted are from 2016 to reflect the time period from which the articles were drawn." Page 6, paragraph 3.

Reviewer 1

I read with interest the manuscript: Lessons to learn on the reporting of AEs in RCTs. A SR. This is a very relevant topic and the authors identified important information that should be voiced. I have a few concerns with the methodology and reporting of findings for this paper and I'll summarize in a few lines here and in more depth below.

Response

We would like to thank the reviewer for their positive comments. We have addressed their comments below and amended the manuscript accordingly.

- 1) I suspect this manuscript is NOT a systematic review (SR). SRs should follow a very systematic method to assure no relevant data has been overlooked. The authors do not report how screening of titles and full papers were done (if in duplicate or not, so I suspect not) and the data collected was not double checked (only 10% of data extracted was checked for accuracy). Also, there is no mention on ROB assessment, it may considered to be omitted in this review, but the authors have to mention the reasons to omit. I believe this paper should be classified as a rapid review based on the above, unless authors can document that many mandatory steps for a SR were indeed followed.

Response

We thank the reviewer for their observation regarding the labelling of this study as a systematic review. We do not believe our research meets the alternative definition as a rapid review i.e. timely information for decision making. The reason we referred to it as a systematic review is because the method to identify eligible literature and the process of evaluation of these papers was systematic. However we acknowledge that the restrictions imposed on the review might mean it does not fully meet the standard definition of a systematic review. Therefore we suggest that we remove the word 'systematic' and simply call it 'a review' and have amended the manuscript throughout.

We note the ambiguity in the manuscript regarding the screening and extraction methods. Preliminary screening was conducted by one reviewer who manually searched the electronic contents table of the journals for reports of original RCTs. Potentially eligible articles were identified based on titles and

abstracts. Any queries were reviewed and discussed with a second reviewer. Whilst formally we only completed a 10% blinded check of extracted data, queries were reviewed by at least two authors and discussed on an ongoing basis. We have clarified this in the methods and discussion section.

We apologise for the oversight regarding risk of bias (ROB). This was included in the PRISMA checklist we submitted alongside the manuscript but we did not discuss it in the body of the manuscript. The authors discussed a ROB assessment and decided it was not relevant because we were not interested in the potential bias of included studies for the efficacy outcomes or drawing inferences from them. We only aimed to describe current practice and not explore current practice by ROB. This has been added to the methods.

Marked revisions

Title: "Analysis and reporting of adverse events in randomised controlled trials; a review"
The word 'systematic' has been removed from the study description throughout the article.

Search strategy

"One reviewer manually searched the electronic contents table of the journals for reports of original RCTs published between September 2015 and September 2016, inclusive. Any queries regarding eligibility were reviewed and discussed with a second reviewer." Page 6/7, paragraph 3.

Data extraction

"A data extraction sheet was piloted and then single data extraction was performed by three reviewers (RP, VC and LH) with 10% independent check of a randomly sampled subset to verify quality. Queries were also discussed between reviewers on an ongoing basis." Page 8, paragraph 2.

Data analysis

"A risk of bias assessment was not undertaken as this study aimed to describe best practice and not evaluate outcomes." Page 8, paragraph 3.

2) In more detailed:

Strengths in limitation box:

- i) Item 1: change to rapid review

Response

We thank the reviewer for their observation regarding the labelling of this study as a systematic review and have addressed this in more detail in comment (1) above.

- ii) Item 2- can you also mention that many weaknesses in methods of reporting were identified and have to be addressed. Not that you only identified good examples.

Response

We thank the reviewer for the suggestion and have added this to the strengths and limitations section.

Marked revisions

"2. This review identifies weakness that need to be addressed as well as good practice that could be adopted." Page 4.

- iii) Item 4- I don't think your review addressed this. Did you specifically review the guidance for best statistical methods to assess causality in RCTs? If you just got a pool of studies from 4 journals and published within 1 year you cannot make such statement.

Response

We thank the reviewer for this comment and have removed this item. Page 4.

3) Objectives:

I suspect the term "best practice" is not accurate. Please use "current practice" or something with more equipoise. Please add that you only looked in RCTs of efficacy. Attention to these changes at the discussion section as well.

Response

We thank the reviewer for this suggestion. We would like to clarify that by 'best practice' we were inferring that we would have expected better practice from these high impact journals as compared to other journals in terms of reporting and analysis of AEs in RCTs. However we acknowledge that this terminology does not accurately reflect this and therefore we have removed all reference to best practice. We have changed this to "current practice in high impact journals" throughout the manuscript.

We have added to the relevant manuscript sections (abstract, introduction, methods and discussion) that we only included studies where the primary outcome was efficacy of the intervention.

Marked revision

Abstract

"To ascertain current approaches to the collection, reporting and analysis of adverse events (AEs) in randomised controlled trials (RCTs) with a primary efficacy outcome." Page 2, Objective.

Introduction

"The aim of this review was to evaluate current practice for collection, reporting and analysis of AEs in RCTs where the primary outcome was efficacy." Page 6, paragraph 2.

Selection criteria

"As the study aimed to assess how authors report and analyse AEs in studies where the primary outcome was efficacy, trials that were specifically designed to investigate safety as a primary outcome were not included." Page 7, paragraph 2.

- 4) 2. Search strategy:
Minor: list journals in alphabetical order

Response

We have amended this accordingly. Page 2, data sources. Page 7, paragraph 2.

5) 3. Methods

- i) 3.1 Screening?
Authors do not report how screening was done. Screening is an important step in a SR and must be clearly reported based on the PRISMA guideline (item 9- study selection) - please report. Screening and data extraction are key factors for a SR, assuring no data is missing.

Response

We note the ambiguity in the manuscript regarding the screening and extraction methods. Preliminary screening was conducted by one reviewer who manually searched the electronic contents table of the journals for reports of original RCTs. Potentially eligible articles were identified based on titles and abstracts. Any queries were reviewed and discussed with a second reviewer. We have checked PRISMA and amended the manuscript so that it is now in line with item 9.

Marked revision

"One reviewer manually searched the electronic contents table of the journals for reports of original RCTs published between September 2015 and September 2016, inclusive. Any queries regarding eligibility were reviewed and discussed with a second reviewer." Page 6/7, paragraph 3.

- ii) 3.2 Data extraction:
Only 10% of data was verified by a second reviewer. I believe since the data extraction could be quite extractor dependent, due to heterogeneity in format of AE reports in RCTs, the data extraction may require some degree of 'judgement call' and it would be more appropriate to have all data checked by a second reviewer, as it is recommended/desirable in a SR. If this step was not followed, I would suggest describing your review as a rapid review, where the reader understands that some main steps form

a SR process were omitted to optimize time and therefore some flaws in the process are expected.

Response

We acknowledge there were restrictions in the review which might not meet the typical definition of a systematic review, therefore we are happy to drop systematic and simply call it a review. We have addressed this point more fully in the response to comment (1) above.

We acknowledge that extraction of data of this kind can be 'extractor dependent'. To ensure accurate data collection there were ongoing informal reviews, where necessary, in addition to the formal independent 10% check. This was to ensure that where there were any doubts over an item authors were in agreement before the extraction was completed. We have clarified this in the manuscript.

Marked response

"A data extraction sheet was piloted and then single data extraction was performed by three reviewers (RP, VC and LH) with 10% independent check of a randomly sampled subset to verify quality. Queries were also informally discussed between reviewers on an ongoing basis. Where specific items were flagged for poor agreement these were re-extracted." Page 8, paragraph 2.

- iii) Please move to results the information that: Agreement between authors was over 80% and please add a little more information on this): How many items had to be re-extracted...

Response

Each reviewer independently extracted data from a randomly selected subset of another reviewers assigned studies. Once this was completed results were compared to establish any discrepancies. A total of 585 items were extracted twice across all three reviewers and a total of 95 discrepancies were identified. This gave agreement of 84%. During this process a number of items were flagged for potential poor agreement. These items were then 100% independently extracted by one author and verified. These included:

- (1) study duration;*
- (2) the AE collection method: passive, prompted, clinical examinations and laboratory measures;*
- (3) timing of collection of prompted and clinical exams and laboratory collection methods;*
- (4) how binary harm outcomes were summarised;*
- (5) whether continuous outcomes were dichotomised;*
- (6) where continuous outcomes were left as continuous how they were analysed.*

Discrepancies related to study duration related to a lack of clarity about the type of data to be extracted with one reviewer incorrectly extracting time of follow-up for primary efficacy endpoint and not total study follow-up. Discrepancies relating to collection method were due to a lack of clarity over the definitions of prompted and passive. The discrepancies between collection of laboratory values and clinical exams were because one reviewer only included this as happening if it explicitly said so, when in fact this could often be discerned from tables of results. Likewise table of results were also indicative if for example laboratory values had been dichotomised but this was often missed. This also impacted the data extracted on analysis methods of continuous outcomes. The discrepancies relating to timing of collection followed from the discrepancies in preceding collection methods questions. These discrepancies were all easily resolved through discussions and amended as appropriate. This information has been added to the results section of the manuscript.

Marked revision

"Data extraction

A total of 585 items were extracted twice across all three reviewers to check the quality of the data extraction. A total of 95 discrepancies were identified. This gave agreement of 84%. During this independent check several items were flagged for potential poor agreement. These items were 100% independently extracted by one author and verified. The items were: study duration; the AE collection method; timing of collection; how binary harm outcomes were summarised; whether continuous outcomes were dichotomised; if continuous outcomes were left as continuous how they were analysed." Page 9, paragraph 2.

- iv) 3.3 Data analysis
Please justify why different methods of data analysis were chosen.

Response

We reported proportions and percentages for binary outcomes and summary statistics including medians, ranges and inter-quartile ranges for continuous outcomes. Details can be found in the data analysis section of the methods. Page 8, paragraph 3.

(1) 4. Results:

Table 3: How AEs info was collected?

- i) The percentages and N do not add up- did studies used more than one methods of AE assessment (active and passive?) if yes, please report it. How many did not report any of these? Please add this info too.
The non applicable item for attribution say that 3 reports included no AE data. What does it mean? No AE were identified/ were not reported/were reported as occurred but no further explanation? Please be clear.

Response

Of the 184 included studies 114 (114/184=62%) reported some form of passive data collection and of these 53 (53/184=28.8%) reported a prompt. The prompted category is subset of the passive group, the percentages have been amended to reflect this. The number of studies that do not report any method of collection is 70 (70/184=38%), this category has been added to the table. Page 12, table 3.

The non-applicable item for attribution reflects 3 studies that included no AE data collection or in fact made any reference to AE data. This has been clarified in the manuscript.

Marked revisions

“Sixty-two percent (n=114) of reports made reference to some form of passive (e.g. spontaneously reported by patients) AE monitoring or collection methods. Of these only 46.5% (53/114) or 29% of total reports included specific details (prompts e.g. questions about specific events or AEs in general, questionnaires, or diaries) regarding these collection methods (Table 2, examples 1-2).” Page 10.

Table 3 Footnote

“3 reports made no reference to AE data throughout the article”. Page 13.

Table 4 Footnote

“3 reports made no reference to AE data throughout the article”. Page 18.

- ii) Selection of AEs and reporting practices-
Here your say that 5 trials did not report any information on AEs and in table 3 you say 3 trials. Please check what is correct.”

Response

The five studies include the three studies that included no information on AE data collection and made no reference to AE data throughout the manuscript, and two studies that made generic statements regarding AEs but provided no details. Hence the figures of 5 and 3 are correct, this has now been clarified in the manuscript.

Marked revision

Selection of AEs and reporting practices

“Two reports only made generic statements regarding AE data: “there were no significant adverse events related to the procedure” and “no excess in mortality or major adverse events were found...”. Three reports made no mention of AEs throughout the manuscript.” Page 13, paragraph 2.

(2) Discussion

- i) In the discussion you have a subtitle called “principle findings” but none of the things reported here were found in your review. Please re-write (or removing the subtitle or adapting the content to our findings).

Response

We thank the reviewer for highlighting this and the subtitle has been removed. Page 21

- ii) Use of CONSORT Harms under principal findings:
You have not specifically checked for the use of CONSORT Harms, so you cannot say that consort harms has not been used based on your findings. If you specifically looked for CONSORT Harms items, please say it earlier in the manuscript (methods and results).

Response

In the review we captured whether or not articles reported the use of CONSORT Harms but we did not assess compliance with the CONSORT Harms statement. We have removed this misleading statement and clarified this section accordingly.

Marked revision

“The CONSORT extension to harm was developed with the aim to improve reporting of safety data in RCTs.³⁷ None of the included studies referenced the CONSTORT HARM extension and of the items in our review that are covered in CONSORT many were not well reported.¹³” Page 21, paragraph 4.

- iii) Maybe I missed, but I did not see if you looked how many journals endorse the CONST Harms. You should not present new information in the discussion (it should belong to results). If I missed the description in results I apologize.

Response

Of the included journals we found that the Lancet was the only journal that made specific reference to the CONSORT Harm extension in their guidelines to authors. We have added this to the results section.

Marked revision

“We assessed any reference to the CONSORT Harm extension and found that none of the included studies mentioned it. Of the four journals included in the review, the Lancet was the only journal that made specific reference to the harm extension in their guidelines to authors.” Page 21, paragraph 1.

- iv) Please add as a limitation that your study was also limited to 2015-2016 and this may not reflect most current practices and previous years.

Response

We have added this to the limitations section.

Marked revisions

“Articles included in this review were published in four of the top ranked medical journals therefore results are likely to be biased towards better findings than we would expect if we included all RCTs and are only for year 2015-2016.” Page 27, paragraph 1.

Reviewer: 2

Thank you for the opportunity to review the manuscript titled “Lessons to learn from the reporting of adverse events in randomised controlled trials: a systematic review”. The manuscript is relatively well written and it presents interesting information about adverse event reporting in RCTs published in four leading medical journals. However, I have several comments/suggestions.

Response

We would like to thank the reviewer for their comments. We have addressed their comments below and amended the manuscript accordingly.

1. The rationale for this review could be made clearer in the Introduction section. The authors appear to cite a number of previous studies documenting the inadequacies of AE reporting AE in journal articles, but it is not made explicitly clear what the gaps in this literature actually are. The first bullet point in the Strengths and Limitations indicates that the unique contribution of the present review is its examination of analysis practices for AEs in RCTs. This seems reasonable, but if so then the present review ought to focus more on this particular aspect. Perhaps this should be reflected in the title as well.

Response

The reviewer rightly highlights that there have been a number of previous studies documenting the inadequacies of AE reporting, however these studies have all been focused on the collection and reporting and none of these studies have looked at analysis practice. This was the main focus of this review. We have changed the title and relevant sections to highlight that analysis practice was the focus for the review.

Marked revisions

Title: "Analysis and reporting of adverse events in randomised controlled trials; a review"

Introduction

"Whilst this work has been undertaken there remains uncertainty about practice for reporting and presenting AE data, and in addition the analysis practice for AEs remains a neglected area for review." Page 6, paragraph 1.

2. There is no strong justification for why only four journals were searched. I understand that the four journals are among the leading general medicine journals, but what was the criteria for selecting journals. Given that only four journals are used as sources, I am not sure it is appropriate to call this a 'systematic review'. At best it is an extremely restrictive systematic review.

Response

We thank the reviewer for their observation regarding the labelling of this study as a systematic review and have addressed this in more detail above in point (1) to the editor and also in point (1) to reviewer 1.

We have provided full justification for only including the four top ranked general medical journals above in point (2) to the editor.

Details of revisions for each of these points can be found with the appropriate responses in the above.

3. Similarly, what is the justification for only including RCTs from the 13-month period between September 2015 to September 2016, inclusive? Why 13 months? Moreover, if the present review is supposed to examine current best practices, then why not include more recent RCTs, say, from 2017?

Response

We chose to survey a one year period and started screening and extraction from October 2016 onwards. Therefore we were unable to include papers from 2017.

Marked revisions

"Articles included in this review were published in four of the top ranked medical journals therefore results are likely to be biased towards better findings than we would expect if we included all RCTs and are only for year 2015-2016." Page 27, paragraph 1.

4. It is unclear whether the study selection process was conducted by one or more independent reviewers. If it was conducted by two or more independent reviewers, then what was the level of agreement between reviewers? And how was potential disagreement resolved?

Response

We note the ambiguity in the manuscript regarding the screening and extraction methods. Preliminary screening was conducted by one reviewer who manually searched the electronic contents table of the journals for reports of original RCTs. Potentially eligible articles were identified based on titles and abstracts. Any queries were reviewed and discussed with a second reviewer. To ensure accurate data collection there were ongoing informal reviews, where necessary, in addition to the formal independent 10% check. This was to ensure that where there were any doubts over an item authors were in agreement before the extraction was completed. We have clarified this in the manuscript.

Marked revisions

Search strategy

“One reviewer manually searched the electronic contents table of the journals for reports of original RCTs published between September 2015 and September 2016, inclusive. Any queries regarding eligibility were reviewed and discussed with a second reviewer.” Page 6/7, paragraph 3.

Data extraction

“A data extraction sheet was piloted and then single data extraction was performed by three reviewers (RP, VC and LH) with 10% independent check of a randomly sampled subset to verify quality. Queries were also discussed between reviewers on an ongoing basis.” Page 8, paragraph 2.

5. In regard to data extraction, the authors state that agreement between reviewers was over 80%. Please state the exact figure. How was the level of agreement measured? Cohen’s kappa? Intraclass correlation coefficient?

Response

The reviewer rightly highlights the lack of detail provided in this section. We have addressed this in detail in comment (5iii) to reviewer 1 and provide details of the amendment to the manuscript.

6. The Results section is unreasonably long (i.e. a whole 10 pages!). As mentioned above, it would be helpful if the authors focused on the unique contribution of their review, that is, the reporting practices of analysis of AEs. (Much of the other information could be relegated to appendices if necessary.)
I would have liked to see a little bit more detail on the reporting practices of AEs.
How many RCTs were powered to detect differences in AEs?
How many inadequately powered RCTs (inappropriately) conducted hypothesis tests of differences in AEs?

Response

We have written the results section in a style as to explain the terminology and concepts as they are presented and to give each data item some context. Our experience is that researchers can be unsure of the importance and purpose of some harm items. Whilst this should ease understanding of the results presented it does mean that the results section is one of the longer sections of the manuscript. An example is given in table 2 which includes exemplars of good practice to give context to the readers. The aim was not to summarise all of the studies included in the review. We would like to retain this table in the article as we think this is useful to demonstrate what good reporting could look like, but it can be moved to the appendix if the reviewers and editor feels this is more appropriate.

The number of pages the reviewer points does include tables embedded within the text so the actual text is only 1344 words.

The full scope of data extracted regarding reporting practices are presented in table 4 and discussed in the results section. This includes information on the analysis population, the unit of analysis e.g.

'patients with at least 1 event', how AE data was summarised e.g. frequencies, percentages, significance tests.

These were all studies with a primary efficacy outcome and safety data presented were either secondary outcomes or unspecified emerging AEs. None of the studies had been powered to detect differences in safety outcomes, therefore any hypothesis testing would be inappropriate. Neither did any of the studies disclose the post-hoc statistical power for detecting the reported differences. We report in the manuscript that "despite a lack of power to undertake formal hypothesis testing, 47% performed such tests for binary outcomes" and we provide an example in the results section: "There were no between-group differences in the rate of patients with at least 1 adverse event (16.7% [14 patients] in the clopidogrel group vs 21.8% [19 patients] in the placebo group; difference, -5.2% [95% CI, -17% to 6.6%]; P = .44)." However with a total safety population of 171 such a test would have only had 13% power to detect such a difference and was therefore substantially underpowered." We did not calculate a post-hoc power test for all studies that performed a hypothesis test on AE data as this was beyond the scope of this review and such a review has already been performed.

Tsang, R., Colley, L., & Lynd, Larry D. 2009 Inadequate statistical power to detect clinically significant differences in adverse event rates in randomized controlled trials. *Journal of Clinical Epidemiology*; 62(6): 609-616. <https://doi.org/10.1016/j.jclinepi.2008.08.005>

https://ac.els-cdn.com/S0895435608002217/1-s2.0-S0895435608002217-main.pdf?_tid=4e659071-3776-47eb-bb4b-2966f9def7ae&acdnat=1538133094_5e4f63bd2d774d0f3fc59263f1d32453

7. The limitations of the present review are understated. Restricting the source of RCTs to four leading medical journals will of course have a major impact on the generalisability of the findings of the present review. But are there no limitations other than the included RCTS being biased toward better results because they were published in top ranking medical journals?

Response

We have added the time period that the articles are taken from as a limitation. We have also added that limiting the independent data check to 10% would not have removed subjectivity from the data extraction but are happy that ongoing informal discussions between authors to clarify any queries would have kept this to a minimum.

Marked revisions

"Articles included in this review were published in four of the top ranked medical journals therefore results are likely to be biased towards better findings than we would expect if we included all RCTs and are only for year 2015-2016. We also acknowledge that only completing 10% independent check of extracted data would not have removed subjectivity from the data extraction but are happy that ongoing discussion between authors to clarify any queries would have kept this to a minimum." Page 27, paragraph 1.

8. One issue I find problematic is that the authors begin with the assumption that RCTs from the four leading medical journals will inform us of current best practice (i.e. lessons to learn), then they proceed with a purely descriptive synthesis of these practices. So far so good. But then the authors turn around, instead of assuming these are best practices, they highlight ways in which these practices are inadequate or inappropriate by pointing to yardsticks such as the CONSORT Statement. At the very least the authors should make it clear up front (i.e. in the Methods section) what criteria or standard the reporting practices are evaluated against.

Response

We agree with the reviewer and have amended the title and relevant sections (aim, strengths and limitations) to reflect that we actually identified weakness that needed to be addressed. However we also highlighted some areas of good practice that could be adopted and have retained this.

Marked revisions

"2. This review identifies weakness that need to be addressed as well as good practice that could be adopted." Page 4.

Reviewer: 3

Overall comments: This review adds to the number of systematic reviews on reporting of adverse events in RCTs. The findings demonstrate that despite the CONSORT harms extension having been published in 2004 there has been no apparent improvement in reporting. In short it is difficult to draw any meaningful conclusions on AEs in evidence from RCTs, be they long term serious outcomes or short term drug related side effects. The review highlights again the key reporting deficiencies and to this extent does not provide any new knowledge, however it confirms the pervasive and persistent nature of the inadequate reporting of AEs.

Response

We'd like to thank the reviewer for their comments. Other research in this area has focused on practices for reporting and presenting AE data. We'd like to highlight that this review goes beyond this adding new knowledge about analysis practice as well as confirming pervasive and persistent practice of inadequate reporting of AEs. Identification of the inadequacies in the analysis of AE data provides researchers an area to focus on with regard to improving practice.

Specific comments:

1. Given that there have been a number of systematic reviews, that have used similar methods for appraisal and synthesis, it is not reasonable to say that this is the first SR to examine and quantify analysis practices for AEs in RCTs.

Response

Other research in this area has focused on practices for reporting and presentation of AE data. None have looked at statistical methods to analyse AE data. We have now clarified this in the introduction and feel it is appropriate to include the statement "this is the first review to examine and quantify AE analysis practice in RCTs published in high impact trials" is accurate.

Marked revision

"Whilst this work has been undertaken there remains uncertainty about practice for reporting, and presenting AE data, and in addition the analysis practice for AE remains a neglected area for review." Page 6, paragraph 1.

2. Publication of an RCT in one of the selected journals does mean that the RCT represents 'best practice'. Rather this would be assessed on the basis of study design and risk of bias which has not been undertaken. Indeed it could be argued that they do not represent best practice due to the deficiencies in reporting of AEs. I agree with the authors that assessment of risk of bias and 'quality' of individual trials is not relevant to the objective of the review. I suggest that reference to best practice be removed. The important point is that the RCTs that have been published over a recent period in examples of high impact general medicine journals are deficient. The limitation of the study is not that these might be 'better' trials, but that the review is limited to four journals and thus provides a 'snap shot' rather than a systematic assessment. A further limitation is that the review is of single publications, whereas AE data may be over a number of reports and would need a systematic review conducted at a study level to capture.

Response

We thank the reviewer for this suggestion. We would like to clarify that by 'best practice' we were inferring that we would have expected better practice from these high impact journals as compared to other journals in terms of reporting and analysis of AEs in RCTs. However we acknowledge that this terminology does not accurately reflect this and therefore we have removed all reference to best

practice. We have changed this to “current practice in high impact journals” throughout the manuscript.

We’d also like to thank the reviewer for the apt summary of the review and have incorporated this into our conclusions.

Marked revisions

Methods

“The top four general medical journals as ranked by impact factors that publish clinical trials of drug interventions were selected: The BMJ (Impact Factor 20.79), the Journal of the American Medical Association (JAMA, IF 44.41), the Lancet (IF 47.8300), and the New England Journal of Medicine (NEJM, IF 72.41). Impact factors quoted are from 2016 to reflect the time period from which the articles were drawn.” Page 6, paragraph 3.

Conclusions and recommendations for future work

“RCTs that have been published over a recent period in examples of high impact general medicine journals are deficient.” Page 28, paragraph 3.

3. Why were trials designed to investigate safety excluded as these may be reports from efficacy studies.

Response

The study aimed to assess how authors report and analyse AEs in studies where the primary outcome was efficacy, therefore trials that were specifically designed to investigate safety as a primary outcome were not included. Studies with a primary safety outcome have a different focus to efficacy studies and as such we would expect different practices, therefore it was not appropriate to incorporate such data into our review.

4. Table 2 could be removed as it does not provide sufficient detail for the reader to be able to form a judgement as to how representative or appropriate or otherwise of the four examples.

Response

Table 2 includes exemplars of good practice to give context to the readers. The aim was not to summarise all of the studies included in the review. We would like to retain this table in the article as we think this is useful to demonstrate what good reporting could look like, but it can be moved to the appendix if the reviewers and editor feels this is more appropriate.

5. The authors should consider structuring the assessment more in line with the 10 domains and 23 items of the CONSORT harms extension. This could be employed as a simple checklist to highlight areas of deficiency as has been done in other reviews with results shown graphically in a manner similar to risk of bias assessments.

Response

There have been many reviews of CONSORT Harms but this was not our aim. This review was primarily to examine what analysis of AEs is undertaken in RCTs. This is where the current gap in knowledge is. We have now clarified the aim throughout the manuscript.

6. The rationale for the components and data items shown in Tables 3 and 4 are not clear this should be expanded in the methodology.

Response

The items extracted from articles and presented in tables 3 and 4 were based on papers by Cornelius et al. and the CONSORT Harm extension paper. These were reviewed and discussed between authors before being piloted and finalised. New items were included based on these discussions to capture information on analysis practices. A rationale for: how AE data was collected (mode of collection, timing) and defined (coding, attribution) during the study; assessment practices of severity of the event or

relatedness to the medicinal product; planned AE analysis (final and interim monitoring plans and analysis populations); how events were selected for inclusion in the journal article; how summary event information was presented in the journal article; and how AEs were analysed are included throughout the discussion. For example: how AE data was collected (mode of collection, timing) and defined (coding, attribution) during the study “has important implications for the type and frequency of AEs reported with “passive collection resulting in fewer recorded AEs”. “The frequency of AE collection has further important implications on the number of events reported. More frequent assessment and longer follow-up will result in more AEs reported.” Page 22, paragraph 2.

Marked revision

“The items to be extracted were based on the work by Cornelius et al. and the CONSORT Harms extension with new items added to capture more specific information on analysis practices. A data extraction sheet was piloted and then single data extraction was performed by three reviewers (RP, VC and LH) with 10% independent check of a randomly sampled subset to verify quality.” Page 8, paragraph 2.

7. The review has not presented an analysis of the types of AEs reported and whether they are ‘important’ or whether they would be expected to be associated with the intervention. For example other reviews have shown a miss-match between known drug related side effects and AEs reported in RCTs. This does not point to a need for research on outcomes, rather a need for reporting to comply with CONSORT. The question of establishing important outcomes e.g. as per COMET is perhaps warranted for other reasons, but this review shows a problem with reporting. The main problem with the current reporting is that when an AE is not reported, it is not known whether it did not occur, was not included in the outcomes and therefore not measured or recorded, or was measured using unknown or inappropriate methodology. In short AE reporting is unreliable.

Response

The reviewer is correct that we have not presented an analysis of the types of AEs reported and whether they are ‘important’. It is very complex to define what is important. Patient quality of life may be deemed of upmost importance to patients but perhaps not from policy makers perspective if considering the cost benefit of a treatment or the impact on clinicians/healthcare providers. Therefore we are supportive of research into core outcomes as per COMET as this would alleviate some of the issues surrounding inconsistent reporting that this and other reviews have highlighted. We also acknowledge the unreliability inherent in AE reporting and suggest in the discussion that the adoption of CONSORT harms by journals may support better reporting. However there are some weaknesses within the CONSORT harms checklist and it is currently undergoing a revision. The new knowledge that will be added to the literature from this review focuses on the inadequacies of analysis practices.

8. The authors could suggest key items of AE reporting that if addressed would improve reliability.

Response

An updated CONSORT Harms is in development and will provide recommendations for authors and so we would not want to replicate/repeat their ongoing work.

Response

This review forms part of an NIHR Doctoral Research Fellowship award that was developed with input from a range of patient representatives. There were no study participants directly involved in the review presented in this manuscript but the original proposal and PPI strategy were reviewed by a service user representative (with experience as a clinical trial participant and PPI advisor) who provided advice, specifically with regard to communication and dissemination to patient and public groups.

Marked revision

“This review forms part of a wider research project that was developed with input from a range of patient representatives. There were no study participants directly involved in this review but the original proposal and PPI strategy were reviewed by service user representatives (with experience as clinical trial participants and PPI advisors) who provided advice, specifically with regard to communication and dissemination to patient and public groups.” Page 8, paragraph 4.

VERSION 2 – REVIEW

REVIEWER	Liliane Zorzela University of Alberta, Canada
REVIEW RETURNED	22-Oct-2018

GENERAL COMMENTS	Thank you for addressing all the comments.
--

REVIEWER	Reidar P. Lystad Macquarie University, Australia
REVIEW RETURNED	24-Oct-2018

GENERAL COMMENTS	Thank you for the opportunity to review the revised manuscript titled “Analysis and reporting of adverse events in randomised controlled trials: a review”. Although the authors have responded adequately to many of the comments raised by the reviewers and editors, some issues remain. See below for specific remaining issues:  1. The rationale for choosing just the top four journals. The authors attempt a partial explanation in their response to the second comment from the Editorial Team, where they mention that their interest was to examine ‘best practice’. The underlying assumption is that best practice is published in the journals with the highest impact factors. Although the assumption is debatable, the main issue is that this assumption is not made explicit to reader. In fact, very little of the reasons outlined in their response to the reviewer comments are actually incorporated into the manuscript. Furthermore, why pick only the top 4 journals? Why not top 3 or top 5, or top 10, or the top quartile? No rationale is offered for this seemingly arbitrary choice. 2. On a related note, the authors state that they want to examine ‘current practice’ and they include publications from the one-year period from September 2015 to September 2016. Data extraction started in October 2016. It is now October 2018. Thus two years have passed and that is arguably a very long time for a systematic review, and more so for a non-systematic review attempting to describe ‘current practice’. What is the reason for it taking two years to complete this review? This review should have been submitted over a year ago to be considered ‘current’. Alternatively, it would have been perfectly feasible to sample RCTs from 2017 and submit the review in mid-2018. 3. In their response to Reviewer #1 (page 8, comment iii), the authors mention that they have added information about The Lancet being the only journal that made specific reference to the CONSORT Extension for Harms in their guidelines to authors. This is useful background information and should probably be
---

	mentioned prior to the Results section (i.e. either in the Introduction or on the Methods). 4. In their response to Reviewer #1 (page 8, comment iv), the authors state that they have added the limitation mentioned by the reviewer to the limitations subsection. That is not actually the case. Although the authors have added a mention of the included RCTs being only from 2015-2016, the authors have omitted stating the limitation. That is, that their sample of RCTs “may not reflect most current practices and previous years”.
--	--

REVIEWER	Martin Howell University of Sydney, Australia
REVIEW RETURNED	22-Oct-2018

GENERAL COMMENTS	In general I am satisfied that the authors have addressed my comments. However I have some qualifications which the authors might like to consider. 1. I still contend that Table 2 could be removed as it does not provide sufficient detail for the reader to be able to form a judgement as to how representative or appropriate or otherwise of the four examples. 2. The authors should consider structuring the assessment more in line with the 10 domains and 23 items of the CONSORT harms extension. This could be employed as a simple checklist to highlight areas of deficiency as has been done in other reviews with results shown graphically in a manner similar to risk of bias assessments. The authors response was dismissive: There have been many reviews of CONSORT Harms but this was not our aim. This review was primarily to examine what analysis of AEs is undertaken in RCTs. This is where the current gap in knowledge is. We have now clarified the aim throughout the manuscript. My response to this is that CONSORT provides the 'gold standard' for reporting and is directly relevant to the stated aim which includes reporting and analysis. Indeed CONSORT identifies the key elements relevant to rationale for selection and analysis of AEs. This is not a review of CONSORT - rather using it as a checklist for AE reporting. 3. The rationale for the components and data items shown in Tables 3 and 4 are not clear this should be expanded in the methodology. I still consider Table 4 not to be fully justified in comparison to using CONSORT as a checklist. This appears to be what Cornelli et al 2013 have done. 4. The authors make reference to revision of CONSORT - however this is not available and not relevant to this paper. The paper would be more constructive if it highlighted key areas of focus that would lead to improvement. This is relevant as the CONSORT extension is large and may not be seen by many trialists as relevant to efficacy trials. Thank you for the opportunity to review the manuscript.
--

VERSION 2 – AUTHOR RESPONSE

We would like to thank the reviewers for their comments and the editorial team for giving us the opportunity to respond and revise our manuscript accordingly.

Reviewer(s)' Comments to Author:

Reviewer: 3

Reviewer Name: Martin Howell

Institution and Country: University of Sydney, Australia Please state any competing interests or state

'None declared': None declared

Please leave your comments for the authors below. In general I am satisfied that the authors have addressed my comments. However I have some qualifications which the authors might like to consider.

1. I still contend that Table 2 could be removed as it does not provide sufficient detail for the reader to be able to form a judgement as to how representative or appropriate or otherwise of the four examples.

Response

Thank you for this confirmation. We are happy to move this table to the supplementary material. And we will highlight in the main text that these exemplars of good practice are available for readers. We feel this is useful to give context to the readers. Supplementary material: table A3, page 6.

Marked revision

We have removed table 2 from the manuscript and added it to the supplementary material as table A3. The table numbers have been updated throughout the manuscript and supplementary material accordingly.

2. The authors should consider structuring the assessment more in line with the 10 domains and 23 items of the CONSORT harms extension. This could be employed as a simple checklist to highlight areas of deficiency as has been done in other reviews with results shown graphically in a manner similar to risk of bias assessments.

The authors response was dismissive: *There have been many reviews of CONSORT Harms but this was not our aim. This review was primarily to examine what analysis of AEs is undertaken in RCTs. This is where the current gap in knowledge is. We have now clarified the aim throughout the manuscript.*

My response to this is that CONSORT provides the 'gold standard' for reporting and is directly relevant to the stated aim which includes reporting and analysis. Indeed CONSORT identifies the key elements relevant to rationale for selection and analysis of AEs. This is not a review of CONSORT - rather using it as a checklist for AE reporting.

Response

We fully accept the reviewers point here and we did not mean to be dismissive but we may not have fully explained our motivation for this work and where this project has come from, which we would like to be considered.

It is already well known that current reporting practice of AEs in clinical trials is inadequate and has been for many years, indeed probably from the advent of clinical trials themselves. We know this from many previous reviews (Reference: 7-16) including reviews of CONSORT harm (Reference: 10, 14-16). We didn't feel the need to replicate this work further.

This review was motivated by the work we are currently undertaking to develop new methods of analysis for AEs, which will be published in the new year. Our primary motivation in this review was to

understand the current practice of analysis of AEs in clinical trials. We undertook this review before we started the methodological work to understand and identify the gaps in analysis practice.

Reporting of AEs in publications, are integral to this aim. We collected additional items not listed in the CONSORT extension to harm that impact on analysis, these included: the definition of the 'safety analysis population', types of data (e.g. binary, continuous) reported, how this data was analysed, use of graphical displays, whether the timing and duration of events was reported, whether multiplicity of testing was accounted for, and whether all events have been reported or only selected events and where these were reported i.e. paper or appendix. We didn't aim to replicate the work that has already shown reporting of CONSORT harm items to be inadequate and as a result did not collect all the items on the list. Please see attached the CONSORT harms checklist and a comparison of what we have collected.

This research identifies and provides evidence for the inefficient use of AE data in clinical trials – both through selective reporting and inadequate analysis. This is subtly different to the aim of assessing poor reporting of AEs by comparing publications to the CONSORT harms. We respectfully request to keep the focus of the paper as it is and not include a CONSORT harm checklist comparison.

3. The rationale for the components and data items shown in Tables 3 and 4 are not clear this should be expanded in the methodology.

I still consider Table 4 not to be fully justified in comparison to using CONSORT as a checklist. This appears to be what Cornelius et al 2013 have done.

Response

Thank you for this comment. We agree these have not been well justified and we have now included additional explanation in the methods section of the manuscript and supplementary material (due to space constraints).

Several items extracted were based on papers by Cornelius et al. and the CONSORT harm extension paper. Additional items were included to capture information on analysis practices so that we could better understand current practice to meet the aim of this review. All these core results are presented in tables 3 and 4 (amended to tables 2 and 3 in updated manuscript).

Marked revision

"The items to be extracted were based on the work by Cornelius et al. and the CONSORT harm extension with additional items added to capture more specific information on analysis practices.^{11, 17} Specifically we focused on the following areas: how AE data was collected (mode of collection, timing) and defined (coding, attribution); how AEs were assessed in terms of severity of the event or relatedness to the medical intervention; if there was any planned AE analysis (final and interim monitoring plans and analysis populations); how events were selected for inclusion in the journal article; how summary event information was presented in the journal article and how AEs were analysed.¹¹ A more detailed rationale for the choice of items extracted is reported in the supplementary material (table A2)." Page 8, paragraph 1.

Supplementary material – page 4

Table A2: Rationale for items extracted

Item	Rationale
How AE data was collected (mode of collection, timing) and defined (coding,	Variation in the collection and definition of events could explain differences in the incidence of observed events. ^{13, 14} For example specifically asking participants about an event of interest in one treatment group whilst relying on patient report in another is likely to lead to a disparity in incidence of events unlikely to be related to the medicinal product.

attribution) during the study.	
Assessment practices of severity of the event or relatedness to the medicinal product.	Attribution of causality by an unblinded assessor allows for subjectivity and bias (even if subconscious) to enter into their decision which can have important implications on the risk-benefit assessment.
Planned AE analysis (final and interim monitoring plans and analysis populations).	For example, the intention-to-treat population is likely to underestimate the AE risk by inflating the denominator. Therefore, this needs to be considered when making conclusions about a drug's safety profile.
How events were selected for inclusion in the journal article.	Due to the space constraints in journal articles it is not always feasible to report all AEs experienced by participants. Therefore, articles often only report a subset of AEs and how these are selected for inclusion has important implications for the safety evaluation. Arbitrary selection criteria can lead to inconsistencies in what is presented across trials for the same disease and/or drug. This prevents an accurate overview of the AEs experienced and invalidates any potential systematic review of events.
How and what summary event information was presented in the journal article.	For example, the number of events and duration of events provides insight into the impact of AEs, with repeated or longer events potentially having far wider clinical implications than a single, shorter event for both patients and prescribers.
How AEs were analysed.	There are many challenges to be considered when analysing AEs in clinical trials. For example, inappropriate statistical testing can lead to misleading conclusions e.g. failure to find a statistically significant result leading authors to conclude that the medicinal product is safe or chance imbalance could lead the authors to erroneously stopping a trial too early. ³⁻⁶

- The authors make reference to revision of CONSORT - however this is not available and not relevant to this paper. The paper would be more constructive if it highlighted key areas of focus that would lead to improvement. This is relevant as the CONSORT extension is large and may not be seen by many trialists as relevant to efficacy trials.

Response

We agree with the reviewer that the CONSORT harm revision should not be referred to as it has not been published yet. We also agree that highlighting key areas of focus would be helpful to the reader and have incorporated suggestions to improve adverse event analysis and reporting in clinical trial report publications into the manuscript.

Marked revision

“Recommendations for consideration for immediate adoption by the clinical trial community are summarised in Table 4.”

Table 4: Recommendations to improve adverse event analysis and reporting in clinical trial report publications

	Recommendation
--	-----------------------

Analysis	Incorporate objective statistical methods to assist the evaluation of adverse event information.
	Consider avoiding dichotomising continuous data.
	When count outcomes are available (such as repeated events within participants) use appropriate statistical methods.
	Clearly define exposure and specify a suitable safety analysis population.
	Use graphical approaches to help summarise large amounts of data.
Reporting	Report adverse event data according to the CONSORT harm checklist.
	Increase the uptake of mandatory submission of CONSORT harm by journals.
	Include a relevant summary of the adverse event profile in the main article. Resist depositing all adverse event data into appendices without summarising.

Pages 26-27.

Reviewer: 1

Reviewer Name: Liliane Zorzela
 Institution and Country: University of Alberta, Canada Please state any competing interests or state 'None declared': None

Please leave your comments for the authors below

Thank you for addressing all the comments.

Reviewer: 2

Reviewer Name: Reidar P. Lystad
 Institution and Country: Macquarie University, Australia Please state any competing interests or state 'None declared': None declared

Please leave your comments for the authors below

Thank you for the opportunity to review the revised manuscript titled "Analysis and reporting of adverse events in randomised controlled trials: a review". Although the authors have responded adequately to many of the comments raised by the reviewers and editors, some issues remain. See below for specific remaining issues:

1. The rationale for choosing just the top four journals. The authors attempt a partial explanation in their response to the second comment from the Editorial Team, where they mention that their interest was to examine 'best practice'. The underlying assumption is that best practice is published in the journals with the highest impact factors. Although the assumption is debatable, the main issue is that this assumption is not made explicit to reader. In fact, very little of the reasons outlined in their response to the reviewer comments are actually incorporated into the manuscript.

Furthermore, why pick only the top 4 journals? Why not top 3 or top 5, or top 10, or the top quartile? No rationale is offered for this seemingly arbitrary choice.

Response

Thank you for highlighting this. We acknowledge that the rationale for including these journals was not made clear to the reader and have now added this to the manuscript.

We also acknowledge that the rationale for including four journals was not included and have added this to the manuscript. An initial scoping review suggested over 100 studies would be eligible from across four journals. This was deemed a feasible number to review given the time and resources available and would provide a sufficient number to evaluate practice.

Marked revision

“The top four general medical journals as ranked by impact factors that publish clinical trials of drug interventions were selected: The BMJ (Impact Factor 20.79), the Journal of the American Medical Association (JAMA, IF 44.41), the Lancet (IF 47.83), and the New England Journal of Medicine (NEJM, IF 72.41). Impact factors quoted are from 2016 to reflect the time period from which the articles were drawn. High impact journals were chosen as we would expect practice in these journals to be of high standard as they include statistical and methodological review. We limited the search to four journals after an initial scoping review revealed around 100 studies would be eligible for inclusion, which was a feasible number to review given the time and resources available and would provide a sufficient number to evaluate practice.” Page 6, paragraph 3.

2. On a related note, the authors state that they want to examine ‘current practice’ and they include publications from the one-year period from September 2015 to September 2016. Data extraction started in October 2016. It is now October 2018. Thus two years have passed and that is arguably a very long time for a systematic review, and more so for a non-systematic review attempting to describe ‘current practice’. What is the reason for it taking two years to complete this review? This review should have been submitted over a year ago to be considered ‘current’. Alternatively, it would have been perfectly feasible to sample RCTs from 2017 and submit the review in mid-2018.

Response

We understand the reviewers point and we have now highlighted this as a limitation in the manuscript. This review took a year to complete and it has been under review for 6 months with BMJ Open. While we acknowledge this as a limitation, given limited progress in the field in nearly twenty years we do not think this delay will mean the review is out of date.

Marked revision

Page 28 paragraph 1

“Articles are only for year 2015-2016 and as such may not reflect the most current practice”.

Page 4 point 3

“Articles included in this review were published in four of the top ranked general medical journals therefore results are likely to be biased towards better practice and are only for year 2015-2016 and as such may not reflect the most current practice.”

3. In their response to Reviewer #1 (page 8, comment iii), the authors mention that they have added information about The Lancet being the only journal that made specific reference to the CONSORT Extension for Harms in their guidelines to authors. This is useful background information and should probably be mentioned prior to the Results section (i.e. either in the Introduction or on the Methods).

Response

Thank you for raising this point. Reviewer one made a similar suggestion in the first round of reviews and asked that we move this information from the discussion to the results section (page 21, paragraph 1). As this information was part of the data extraction process we feel that it more rightly

belongs in the results section. However, if the editor deems that it would be better suited in the introduction we are happy to move it.

4. In their response to Reviewer #1 (page 8, comment iv), the authors state that they have added the limitation mentioned by the reviewer to the limitations subsection. That is not actually the case. Although the authors have added a mention of the included RCTs being only from 2015-2016, the authors have omitted stating the limitation. That is, that their sample of RCTs “may not reflect most current practices and previous years”.

Response

Thank you for highlighting this. We have now explicitly stated this as a limitation in the discussion and strengths and limitations section of the manuscript.

Marked revision

Page 28 paragraph 1

“Articles are only for year 2015-2016 and as such may not reflect the most current practice”.

Page 4 point 3

“Articles included in this review were published in four of the top ranked general medical journals therefore results are likely to be biased towards better practice and are only for year 2015-2016 and as such may not reflect the most current practice.”

VERSION 3 – REVIEW

REVIEWER	Reidar P. Lystad Macquarie University, Australia
REVIEW RETURNED	22-Dec-2018
GENERAL COMMENTS	Thank you for the opportunity to re-review the manuscript titled "Analysis and reporting of adverse events in randomised controlled trials: a review". The authors have satisfactorily addressed my comments.

VERSION 3 – AUTHOR RESPONSE

We would like to thank the reviewer for their comments throughout this process and their recommendation to publish. We'd also like to thank the editorial team for the additional comments which we address below and for giving us the opportunity to respond and revise our manuscript accordingly.

Editors' comments:

1. Given that the search has not been updated (findings would likely be similar) it may be more appropriate to say “contemporary practice” rather than “current practice”.

Response

Thank you for this comment. We agree that the phrase “contemporary practice” is more appropriate than “current practice” when describing the aim of this project and have amended the manuscript accordingly. We have retained the use of “current practice” when describing the limitations of the project.

Marked revision

Abstract - Objective Page 1

“To ascertain contemporary approaches to the collection, reporting and analysis of adverse events (AEs) in randomised controlled trials (RCTs) with a primary efficacy outcome.”

Strengths and limitations of the study – Page 4

“3. Articles included in this review were published in four of the top ranked general medical journals therefore results are likely to be biased towards better practice

4. Included articles are only for year 2015-2016 and as such may not reflect current practices.”

Introduction – Page 5, paragraph 1

“However, contemporary analysis and reporting practices are inadequate.”

Page 6, paragraph 2

“The aim of this review was to evaluate contemporary practice for collection, reporting and analysis of AEs in RCTs where the primary outcome was efficacy.”

Discussion – page 19, paragraph 5

“The aim of this study was to review contemporary practice across four leading medical journals for AE collection, analysis and reporting practices, highlighting any areas for improvement and examples of good practice.”

Page 26, paragraph 1

“Articles are only for year 2015-2016 and as such may not reflect current practice.”

-
2. Please include a summary of the findings in the first paragraph of the discussion and perhaps take out the last sentence of the conclusions, namely “RCTs that have been published over a recent period in examples of high impact general medicine journals are deficient” (which can be misinterpreted), and rewrite the conclusions to read something like “(...) Analysis of AE data is frequently inappropriate and reports such as RCTs that have been published over a recent period in high impact general medical journals often provide insufficient and inconsistent information to allow a comprehensive summary of the safety profile to be established”.

Response

We thank the editor for their comment and have now included a summary of our findings at the beginning of the discussion (page 20, paragraph 1). We have also amended the concluding paragraph (page 26, paragraph 2) as per their suggestion.

Marked revisions

Page 20, paragraph 1

“We found that the collection, reporting and analysis of AE data in clinical trials is inconsistent and RCTs as a source of safety data are underutilised. Analysis of AE was often inappropriate with suboptimal practice including ignoring valuable information on repeated events and inappropriate practice of underpowered multiple hypothesis testing.”

Page 26, paragraph 2

“Analysis of AE data is frequently inappropriate and RCT reports published over a recent period in high impact general medical journals often provide insufficient and inconsistent information to allow a comprehensive summary of the safety profile to be established.”

3. Please revise points 1 and 2 of the strengths and limitations to refer specifically to methodological points.

Response

Thank you for this comment. We have amended points 1 and 2 accordingly. We have also split point 3 into two items.

Marked revision

Strengths and limitations of this study – page 4

- “1. This is the first review to examine and quantify the methods used for AE analysis in RCTs published in high impact general medical journals.
2. This review identifies methodological weakness that need to be addressed as well as good practice that could be adopted.
3. Articles included in this review were published in four of the top ranked general medical journals therefore results are likely to be biased towards better practice.
4. Included articles are only for year 2015-2016 and as such may not reflect current practice.” Reviewer(s)' Comments to Author:

Reviewer: 2

Reviewer Name: Reidar P. Lystad

Institution and Country: Macquarie University, Australia Please state any competing interests or state 'None declared': None declared

Please leave your comments for the authors below Thank you for the opportunity to re-review the manuscript titled "Analysis and reporting of adverse events in randomised controlled trials: a review". The authors have satisfactorily addressed my comments.

Response

We would like to thank the reviewer for their comments throughout this process and their recommendation to publish.